# On the Moderating Effects of Country Governance on the Relationships between Corporate Governance and Firm Performance

Chiu-Hui Wu 



Department of Finance, Shih Hsin University, Taipei City 116, Taiwan; chiuhui@mail.shu.edu.tw

**Abstract:** This study further empirically examines the influence of board independence on financial performance by using the world's top 1000 firms. CEO duality and the percentage of independent directors are used as the indicators of board independence. Moreover, this study re-addresses the findings in the literature by giving supplement in theory and conducting tests for the influence of board independence on firm performance as well as the moderating effects of country governance, focusing on regulatory quality and rule of law, with multi-level modeling, a more sophisticated statistical approach. Four hypotheses, based on agency theory and compensation theory, were developed. The results indicated that CEO duality and the percentage of independent directors exerted, respectively, negative and positive influence on Return on Assets (ROA), a firm performance indicator. Furthermore, regulatory quality and the rule of law positively moderated the negative effects of the former and negatively moderated the positive effects of the latter. Some practical implications were discussed based on the results obtained.

**Keywords:** corporate governance; country governance; firm performance; multilevel modeling

## 1. Introduction

Corporate governance has been receiving much attention. Valukas' (2010) report on Lehman Brothers' efforts to mask its debt is disheartening to anyone who thought that Enron Corporation represented the worst of financial statement manipulation and hoped that the Sarbanes–Oxley Act of 2002 would remedy the problem. Furthermore, WorldCom overstated its income and balance sheet and through its misstatements created a loss for shareholders of as much as USD 200 billion in 2002. The accounting calamity at Italian dairy-foods giant Parmalat has been one of the largest financial frauds in history. A lesson I can learn from the above is that companies should emphasize the importance of business ethics and corporate governance. Corporate governance deficiencies can even lead to financial crises (Claessens and Yurtoglu 2013). Business ethics and corporate governance positively impact firm performance (Erondu et al. 2004). Analyzing the change in corporate governance is important for firm valuation (Morey et al. 2009). After serious corporate governance and accounting problems at several prominent companies such as Enron, HealthSouth, Worldcom, Alstom, Parmalar, and many others came to light at the start of the new millennium, there has been an explosion of sorts in research on and practice of corporate governance. The World Bank itself had formed a corporate governance forum in 1999 to improve governance practices. Indeed, a variety of empirical studies have looked at this question. A large number of research reports on corporate governance were published after the Enron crisis. Corporate governance has become a big concern in financial crisis prediction. The global financial crisis has challenged conventional thinking on state ownership of financial institutions and forced policymakers to reconsider the role

that the structure of financial systems plays, especially state ownership of firms and other forms of corporate governance.

Corporate governance mechanisms ensure stakeholders in corporations get a return on their investments (Shleifer and Vishny 1997). Corporate governance varies widely across firms, countries, and the globe. Numerous studies have shown that differences in legal institutions explain much of the cross-country differences in corporate governance through ownership (La Porta et al. 2002; Harford et al. 2012; Calomiris and Carlson 2016). Black et al. (2006) examine many detailed corporate governance scorecards in Korea and find that the incremental explanatory power of firm-specific characteristics is similar. A professional corporate governance culture can capture the motivation and learning strategies and provide prospective investors with enhanced returns on investments. Lukas and Basuki (2015) provide a literature review for corporate governance mechanisms, financial performance, and the relationship between corporate governance and financial performance. If the nature and magnitude of more subtle corporate governance activities can be captured, more insightful information can be obtained.

La Porta et al. (2000) find that Korean firms with low ownership concentration show low firm profitability, controlling for firm and industry characteristics, during 1993–1997. Calomiris and Carlson (2016) link different managerial ownerships to different corporate governance policies, attitudes toward risk, and methods of risk management. They find that formal corporate governance and high manager ownership are negatively correlated with firm performance. Mitton (2002) shows that corporate governance measures, such as high disclosure quality and concentrated ownership, affect the stock market valuations of firms during periods of crisis. Bhagat and Bolton (2008) mention that efforts to improve corporate governance should focus on the stock ownership of board members. Harford et al. (2012) measure insider ownership as the ratio of top-five insider holdings of common stocks to the total shares outstanding, and institutional ownership as the ratio of shares that institutions own in a firm divided by the total number of shares outstanding. Corporate governance is important to build the foundations for the relationship between the directors, board of directors, and shareholders which leads to clarity in the rights and obligations of each party (Donnelly and Mulcahy 2008; Forker 2012; Barros et al. 2013; Crifo et al. 2015). Hallock (1997) and Bhagat and Bolton (2013) emphasize the role of networks among CEOs that serve on boards and the adverse impact on the governance of such firms.

The importance of corporate governance for firm valuation, firm performance, and firm stability has been acknowledged (Bhagat and Bolton 2008; Morey et al. 2009). Korean rules impose special governance requirements on large firms and firm risk (Black et al. 2006). Using a sample of 597 French listed firms during 2001–2007, Boubaker et al. (2015) find that firms appointing independent directors and splitting CEO and the chairman of the board accumulate fewer cash reserves. Belkhir et al. (2014) investigate the effects of the separation of control and ownership on the value of cash holdings in publicly listed French firms and find that board independence is basically reflected in better market valuation of corporate cash holdings. Lemmon and Lins (2003) study the effect of ownership structure on value during the region's financial crisis. Factors such as board structure (Vafeas and Theodorou 1998; Bonn 2004; García-Meca et al. 2015; Koji et al. 2020), board gender diversity (Boubaker et al. 2014; Liu et al. 2014), board independence (Hermalin and Weisbach 1998; Joseph et al. 2014), stock ownership of board members (Bhagat and Bolton 2013), and whether the Chairman and CEO positions are occupied by the same or two different individuals (Brickley et al. 1997; Bhagat and Bolton 2008) have also been examined. Governance proxy variables include managerial ownership (e.g., Calomiris and Carlson 2016), firm's insider shareholdings (Gorton and Rosen 1995; Harford et al. 2012), ownership concentration (La Porta et al. 2000), and the ownership percentage of the single largest shareholder (Beltratti and Stulz 2012), and multiple directorships and types of shareholders (Doidge et al. 2007; Kasipillai and Mahenthiran 2013), the number of independent directors, the proportion of outside directors (other than corporate insiders) and whether CEO is

the chairman of the board as indicators of governance (Brickley et al. 1997; Boubaker et al. 2015), and board size (Yermack 1996; Hermalin and Weisbach 2003). A focus of corporate governance is board independence (Duru et al. 2016; Mukaddam and Athenia 2020) and is adopted in this study.

Country governance is important. Ngobo and Fouda (2012) indicate that good country governance can improve firm performance when it achieves the rule of law and reduces corruption. Country governance consists of six dimensions: voice and accountability, political stability and absence of violence/terrorism, government effectiveness, regulatory quality, rule of law, and control of corruption (Kaufmann et al. 2009). Countries with good governance, such as the rule of law and specific legal protection for investors and creditors, will be able to develop large and liquid financial systems (La Porta et al. 1998). The rule of law reflects judicial integrity and respect for property rights. The rule of law measures the extent to which agents have confidence in and abide by the rules of society. A higher score for rule of law implies more confidence in the legal system (Kaufmann et al. 2009). Essen et al. (2013) find that firms located in countries with more developed legal frameworks perform better during a financial crisis. Good country governance can improve firm performance when it achieves the rule of law and reduces corruption (Ngobo and Fouda 2012). A consistent and efficient country governance mechanism can reduce the risks in financial markets that are detrimental to economic growth. Bruno and Claessens (2010) investigate how company-level corporate governance practices and country-level legal investor protection jointly affect firm performance.

Specifically, there are two research objectives of this study. First, this study further empirically examines the influence of board independence on financial performance by using the world's top 1000 firms. CEO duality (meaning that the CEO plays the role of board chair also) and the percentage of independent directors are used as the indicators of board independence. Second, this study re-addresses the findings of Bruno and Claessens (2010) by giving supplement in theory and conducting tests for the influence of board independence on firm performance as well as the moderating effects of country governance, focusing on regulatory quality and rule of law, with multi-level modeling, a more sophisticated statistical approach.

The rest of the article is organized as follows. The next section provides a discussion on the relevant literature and develops research hypotheses. Section 3 presents the methodology. Empirical results and robustness checks are presented in Section 4. Finally, Section 5 discusses implications.

## 2. Hypothesis Development

### 2.1. Board Independence and Firm Performance

Recent studies have focused on examining whether corporate governance has effects on the quality of financial performance. Clacher et al. (2008) examine the effect of corporate governance on two performance measures, Tobin's Q and ROA. They indicate that compliance with governance practices improves firm value and performance. Zagorchev and Gao (2015) examine the effects of corporate governance for financial institutions in the US between 2002 and 2009. They find that governance is positively related to US banks' performance. Choi et al. (2007) empirically show that there was a positive relationship between board independence and firm performance in South Korea during the Asian crisis. Ferris and Yan (2007) find that firm performance is positively related to board independence. Bruno and Claessens (2010) find that the presence and independence of corporate governance has a positive impact on firm value. Duru et al. (2016) provide evidence that CEO duality has a statistically significant negative impact on firm performance that is positively and significantly moderated by the percentage of independent directors.

The situation where the CEO chairs the group of people in charge of monitoring and evaluating the CEO's performance, known as "CEO duality", is problematic from an agency perspective. Dayton (1984) and Levy (1981) suggest that CEO duality diminishes the monitoring role of the board of directors over the executive manager, and in turn

exerts a negative effect on corporate performance. Separation of the CEO and the board of directors will improve firm performance because the board can better monitor the CEO (Harris and Helfat 1998). According to agency theory, the separation of ownership and control, typical of large corporations, can be an efficient form of economic organization (Jensen and Meckling 1976; Fama 1980; Fama and Jensen 1983; Dey 2008).

Berg and Smith (1978) used data from Fortune 200 firms and reported a negative relationship of CEO duality with return on investment (ROI). Duru et al. (2016) indicate that CEO duality has a significantly negative impact on firm performance. CEO duality might reduce firm performance through managerial entrenchment. Arora and Sharma (2016) point out that CEO duality is associated with inefficiency and lower firm performance. The agency theory proposes that CEO duality and firm performance are negatively related. When ownership and management are separated, efficiencies may arise from the organization (Westhead and Howorth 2006). Combining the positions of CEO and board chairman weakens control, and negatively affects firm performance.

**Hypothesis 1 (H1).** *CEO duality is negatively related to firm performance.*

Johnson et al. (1996) have indicated that independent directors play a more effective and efficient monitoring role than executive directors. Thus, independent directors play a more crucial role in monitoring the financing issues of companies. Independent directors may exercise better control, reduce agency costs, bring outside resources, and increase financial transparency.

Some empirical evidence suggests that outside directors can improve board effectiveness and firm performance. By using 229 Australian firms, Hutchinson (2002) finds that more independent directors on the board lead to better positive financial performance of the firm. Peng (2004) indicates that institutional outside directors have a positive impact on firm performance in China. Luan and Tang (2007) suggest that independent outside director appointments do have a significantly positive impact on a firm's performance after controlling for a firm's past performance. Choi et al. (2007) report that the effects of independent outside directors on firm performance are strongly positive. Masulis and Mobbs (2014) indicate that the higher intensity of board monitoring on the CEO leads to better firm performance. The Sarbanes–Oxley Act (SOX) of 2002 that introduced corporate governance suggests independent directors can mitigate the agency conflict between management and shareholders. Independent directors play an active and important role in the overall board.

**Hypothesis 2 (H2).** *The percentage of independent directors is positively related to firm performance.*

*2.2. Country Governance, Corporate Governance, and Firm Performance*

A legal system, including laws, rules, and regulations, is one of the most important attributes of a country's governance infrastructure (Sun et al. 2015). Thus, the dimensions of regulatory quality and rule of law of country governance can reflect the legal environment and is the focus of this study. As reviewed in the previous section, legal environment is positively associated with firm performance. If a country provides a better legal environment, then companies in that country would benefit more. Companies located in countries with better governance tend to perform better, so corporate governance may exert less influence on firm performance. Although corporate governance is positively related to firm performance, the relationship is weaker for companies located in countries with better legal environments. In contrast, companies located in countries with poor legal environments are more likely to perform worse. Under the circumstances, improving corporate governance becomes a particularly effective way to strengthen firm performance. Corporate governance would be more valuable for companies in a poor legal environment. According to the compensation theory (Champoux 1978; Staines 1980), corporate governance compensates for the deficiency of country governance. Thus, country governance (regulatory quality and rule of law) positively moderates the negative relationship between CEO duality and

firm performance (ROA) and negatively moderates the positive relationship between the percentage of independent directors and ROA. Specifically, I hypothesize:

**Hypothesis 3 (H3).** *Country governance moderates the effects of CEO duality on firm performance in such a way that the negative relationship between CEO duality and firm performance is weaker for companies in countries with higher levels of governance.*

**Hypothesis 4 (H4).** *Country governance moderates the effects of the percentage of independent directors on firm performance in such a way that the positive relationship between the percentage of independent directors and firm performance is weaker for companies in countries with higher levels of governance.*

A research framework showing the four research hypotheses is depicted in Figure 1.

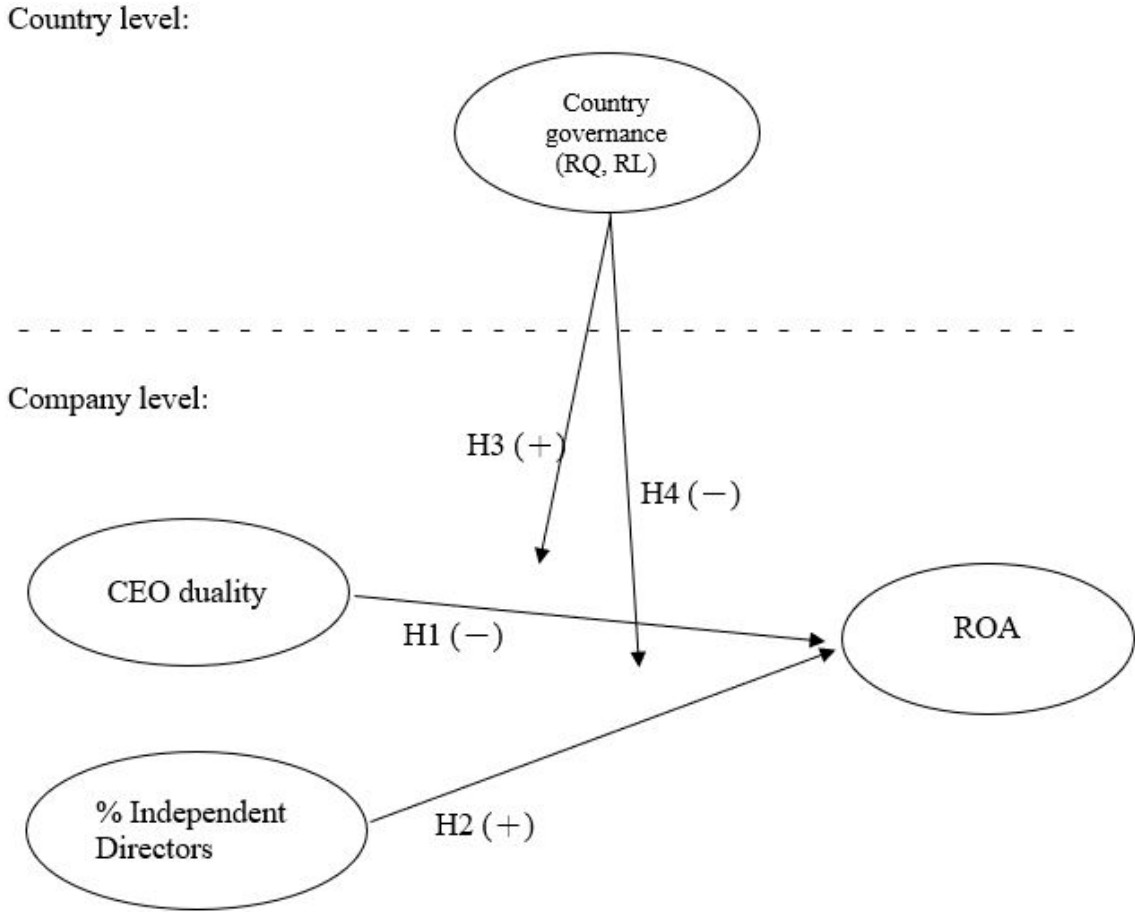

**Figure 1.** Research framework showing hypothesized relationships. Note: RQ = Regulatory quality; RL = Rule of law.

### 3. Methods

#### 3.1. Measures and Data

ROA and Tobin's Q are widely used indicators of firm performance in corporate governance research (e.g., Clacher et al. 2008; Bruno and Claessens 2010; García-Meca et al. 2015). In this study, ROA was used as the proxy of firm performance. ROA is net income after tax as a percentage of the average book value of total assets. CEO duality and the percentage of independent directors are used as the indicators of board independence. CEO duality indicates that the company's chief executive officer is also chairman of the board. The percentage of independent directors is the proportion of independent directors on the board. Kaufmann et al. (2010) developed and maintained a country governance

database covering over 200 countries. The score data for the dimensions of regulatory quality and rule of law are available from the Kaufmann et al. database. The data for CEO duality, the percentage of independent directors, and firm performance were collected from the Bloomberg database (a comprehensive financial database for individual companies in the world). The time period was from 2009 through 2018. The control variables used include firm size, sales growth, and time (Bruno and Claessens 2010). Definitions of all variables used and source of data are shown in Table 1.

**Table 1.** Definitions of variables.

| Variable | Source of Data | Definition |
|---|---|---|
| **Country Governance** | | |
| Regulatory quality (RQ) | Kaufmann et al. (2010) | Measures ability of the government to formulate and implement sound policies and regulations that permit and promote private sector development |
| Rule of Law (RL) | Durnev and Kim (2005) | Measures the extent to which "legal" is the product of anti-director and rule of law, although our measure is constructed using the updated version of anti-rights |
| **Corporate governance** | | |
| CEO duality (CEOD) | Bloomberg | The Chairman of the Board of Directors of the same firm |
| Percentage of independent directors (PID) | Bloomberg | Percentage of independent directors to the number of board directors |
| **Firm Performance** | | |
| ROA | Bloomberg | Return on assets, the net income after tax as a percentage of the average book value of total assets |
| **Firm Size** | | |
| ASSET (USD million) | Bloomberg | Logarithm of total assets |
| **Sales Growth** | | |
| SG | Bloomberg | One-year growth of sales |

The Bloomberg database may not provide full financial information of companies. So I further collected supplemental data from financial statements and reports from financial regulatory agencies of different countries. At the same time, financial yearbooks and individual firm financial statements were used to double-check the data. The final dataset with complete information includes 830 companies (out of the world's top 1000) across 43 countries from 2009 to 2018. The data consist of three levels: time level, company level, and country level. Time is nested within the company, which is nested within the country. A sample frequency distribution is presented in Table 2.

*3.2. Multi-Level Modeling*

Based on the data structure mentioned above, three-level modeling was used for the analysis. Singer (1998) and Peugh and Heck (2017) illustrated fitting three-level models by using SAS/PROC MIXED, which was used in this study too. There are three sources of firm performance variation: (a) variation in firm performance across 10 repeated measurements (Level 1) for 830 companies nested in 43 countries; (b) variation in the average firm performance across all companies (Level 2) nested in 43 countries; and (c) variation in the average firm performance across all countries (Level 3). Predictors are centered on their means to facilitate interpretation. Let $ROA_{ijt}$ be the measured performance, $CEOD_{ijt}$ and $PID_{ijt}$ be the two corporate governance indicators, $TA_{ijt}$ be the logarithm of total assets (a proxy of firm size, serving as a control variable), and $SG_{ijt}$ be one-year growth of sales (serving as another control variable) at time $t$ (Level 1) for company $j$ (Level 2) in country $i$ (Level 3). Let $\overline{TA}_{ij}$, $\overline{SG}_{ij}$, $\overline{Time}$, $\overline{CCEOD}_{ij}$, and $\overline{CPID}_{ij}$ denote the mean values of $TA_{ijt}$, $SG_{ijt}$, $Time_t$, $CEOD_{ijt}$, and $PID_{ijt}$ across 10 repeated measurements, and $CTA_{ijt}$ (= $TA_{ijt} - \overline{TA}_{ij}$),

$CSG_{ijt}$ ($= SG_{ijt} - \overline{SG}_{ij}$), $CTime_t$ ($= Time_t - \overline{Time}$), $CCEOD_{ijt}$ ($= CCEOD_{ijt} - \overline{CCEOD}_{ij}$), and $CPID_{ijt}$ ($= CPID_{ijt} - \overline{CPID}_{ij}$) denote the centered $TA_{ijt}$, $SG_{ijt}$, $Time_t$, $CEOD_{ijt}$, and $PID_{ijt}$. The three-level model (Model 1) used to test for Hypotheses 1 and 2 (the main effects of CEO duality and the percentage of independent directors) is given by

$$ROA_{ijt} = \gamma_{00} + \beta_{01}CTA_{ijt} + \beta_{02}CSG_{ijt} + \beta_{03}CTime_t + \gamma_{10}CCEOD_{ijt} + \gamma_{20}CPID_{ijt} + (u_{0i} + r_{0ij} + \varepsilon_{ijt}), \quad (1)$$

where, $i = 1, 2, \ldots, 43$; $j = 1, 2, \ldots, 830$; $t = 1, 2, \ldots, 10$, $\beta_{01}, \beta_{02}$, and $\beta_{03}$ represent, respectively, the fixed effects associated with the control variables of *TA*, *SG*, and *Time*, $\gamma_{10}$ and $\gamma_{20}$ denote, respectively, the fixed effects associated with *CEOD* and *PID*. $\gamma_{00}$ is the mean of *ROA* across countries when *CTA*, *CSG*, *CTime*, *CCEOD*, and *CPID* are zero, and $\varepsilon_{ijt}$, $r_{0ij}$, and $u_{0i}$ denote, respectively, Level 1, 2, and 3 errors, with variances $\sigma_\varepsilon^2$, $\sigma_r^2$, and $\sigma_u^2$. $\varepsilon_{ijt}$ is the deviation from the mean performance across repeated measurements for company $j$ nested in country $i$. $r_{0ij}$ is the deviation from the mean performance across companies nested in country $i$. $u_{0i}$ is the deviation from the mean performance across countries. $\varepsilon_{ijt}$, $r_{0ij}$, and $u_{0i}$ are assumed to be uncorrelated. However, $\varepsilon_{ijt}$'s tend to be autocorrelated. Their covariance structure $\mathbf{\Theta}_\varepsilon$ is assumed to be identical for all companies. A commonly used covariance structure is based on AR(1) (the first-order autoregressive) (Littell et al. 2006, p. 175), and is given by

$$\mathbf{\Theta}_\varepsilon = \sigma_\varepsilon^2 \begin{bmatrix} 1 & \rho & \cdots & \rho^9 \\ & 1 & \cdots & \rho^8 \\ & & \ddots & \vdots \\ & & & 1 \end{bmatrix}, \quad (2)$$

where, $\rho$ is the autoregressive coefficient (as well as the autocorrelation coefficient) for the process of $\varepsilon_{ijt} = \rho\varepsilon_{ij,t-1} + v_{ijt}$, where $v_{ijt}$ denotes an i.i.d. disturbance process.

In order to test for Hypotheses 3 and 4, a country governance indicator and its cross-products with *CEOD* and *PID* were added to Model 1. Two indicators of country governance, RQ and RL, were used in this study for the purpose of robustness check. Let $\overline{RQ}$ and $\overline{RL}$ denote the means of *RQ* and *RL* across countries, respectively, and *CRQ* ($= RQ_i - \overline{RQ}$) and *CRL* ($= RL_i - \overline{RL}$) the centered *RQ* and *RL* for country $i$. The three-level model (Model 2) used to test for Hypotheses 3 and 4 with the indicator of *RQ* (the moderating effects of *RQ* on the influences of CEO duality and the percentage of independent directors on *ROA*) is given by

$$\begin{aligned} ROA_{ijt} = {}& \gamma_{00} + \beta_{01}CTA_{ijt} + \beta_{02}CSG_{ijt} + \beta_{03}CTime_t + \gamma_{10}CCEOD_{ijt} + \gamma_{20}CPID_{ijt} + \gamma_{01}CRQ_i + \\ & \gamma_{11}CCEOD_{ijt} \times CRQ_i + \gamma_{21}CPID_{ijt} \times CRQ_i + (u_{0i} + r_{0ij} + \varepsilon_{ijt}), \end{aligned} \quad (3)$$

where, $\gamma_{11}$ and $\gamma_{21}$ represent the moderating effects of country governance indicator *RQ* on the relationships between corporate governance indicators *CEOD* and *PID* and ROA, respectively, after controlling for firm size, sales growth, and time. The three-level model (Model 3) used to test for Hypotheses 3 and 4 with the indicator of *RL* is similar to Model 2, by using *CRL* instead of *CRQ*, and is given by

$$\begin{aligned} ROA_{ijt} = {}& \gamma_{00} + \beta_{01}CTA_{ijt} + \beta_{02}CSG_{ijt} + \beta_{03}CTime_t + \gamma_{10}CCEOD_{ijt} + \gamma_{20}CPID_{ijt} + \gamma_{02}CRL_i + \\ & \gamma_{11}CCEOD_{ijt} \times CRL_i + \gamma_{21}CPID_{ijt} \times CRL_i + (u_{0i} + r_{0ij} + \varepsilon_{ijt}), \end{aligned} \quad (4)$$

where $\gamma_{11}$ and $\gamma_{21}$ become the moderating effects of the country governance indicator *RL*. Note that the tests associated with *CCEOD* and *CPID* in Model 1 and those associated with the cross-product terms in Models 2 and 3 are one-tailed.

**Table 2.** Sample frequency distribution.

| Country | Frequency | Country | Frequency |
|---|---|---|---|
| Argentina | 1 | Macau | 1 |
| Australia | 12 | Malaysia | 3 |
| Austria | 2 | Mexico | 4 |
| Belgium | 3 | Netherlands | 13 |
| Bermuda | 1 | Norway | 3 |
| Brazil | 11 | Peru | 2 |
| Canada | 32 | Philippines | 2 |
| China | 87 | Qatar | 2 |
| Colombia | 1 | Russia | 9 |
| Denmark | 4 | Saudi Arabia | 6 |
| Finland | 5 | Singapore | 5 |
| France | 31 | South Africa | 4 |
| Germany | 25 | South Korea | 10 |
| Hong Kong | 22 | Spain | 7 |
| India | 25 | Sweden | 13 |
| Indonesia | 6 | Switzerland | 24 |
| Ireland | 6 | Taiwan | 9 |
| Israel | 1 | Thailand | 5 |
| Italy | 7 | United Arab Emirates | 3 |
| Japan | 68 | United Kingdom | 37 |
| Kuwait | 1 | United States | 316 |
|  |  | Vietnam | 1 |

## 4. Results

### 4.1. Hypotheses Tests

Results for the three-level modeling are reported in Table 3. Models 1 and 2 are the models given in Equations (1) and (3), respectively. In Model 1, the effect of CEO duality on ROA was negative and significant ($\hat{\gamma}_{10} = -0.0186$, $p < 0.001$), and the effect of the percentage of independent directors on ROA was positive and significant ($\hat{\gamma}_{10} = 0.0224$, $p = 0.0036$), providing support for Hypotheses 1 and 2. These results show the negative relationship between CEO duality and firm performance, consistent with prior studies (e.g., Arora and Sharma 2016) which indicate that CEO duality is associated with inefficiency and lower firm performance. The finding that CEO duality weakens firm performance supports the agency theory, which argues that CEO duality creates ambiguous leadership decisions and problems. Moreover, the results show the positive impact of the percentage of independent directors on firm performance, confirming previous studies (e.g., Arora and Sharma 2016) indicating that more independent directors lead to more financial transparency and better firm performance. On the contrary, deficiency of board independence reflects governance problems and will lower firm performance.

In Model 2, I found that the quality of the regulatory mechanism positively moderates the former effect ($\hat{\gamma}_{11} = 0.0046$, $p = 0.035$), and negatively moderates the latter ($\hat{\gamma}_{21} = -0.0192$, $p = 0.0481$), providing support for Hypotheses 3 and 4. The negative effect of CEO duality and the positive effect of the percentage of independent directors on firm performance would be partly offset by country governance, implying that the effects of corporate governance on firm performance depend on how country governance performs. The positive influence of corporate governance is weakened when country governance is adequate but strengthened when country governance is inadequate. The results confirm that corporate governance can compensate for the deficiency of country governance.

**Table 3.** Results of three-level modeling.

|  | Model 1 | Model 2 | Model 3 |
|---|---|---|---|
| Intercept | 6.6145 *** | 6.2525 *** | 6.3168 *** |
| *CTA* | −0.4274 * | −0.4342 * | −0.4353 * |
| *CSG* | 0.0018 *** | 0.0018 *** | 0.0018 *** |
| *CTime* | 0.1328 *** | 0.1192 *** | 0.1286 *** |
| *CCEOD* | −0.0186 *** | −0.0201 *** | −0.0200 *** |
| *CPID* | 0.0224 ** | 0.0270 ** | 0.0297 ** |
| *CRQ* |  | 0. 8037 ** |  |
| *CCEOD* × *CRQ* |  | 0.0046 * |  |
| *CPID* × *CRQ* |  | −0.0192 * |  |
| *CRL* |  |  | 0.6581 * |
| *CCEOD* × *CRL* |  |  | 0.0029 + |
| *CPID* × *CRL* |  |  | −0.0186 * |
| Variance estimate |  |  |  |
| $\hat{\sigma}_u^2$ | 1.0376 | 0.168 | 0.442 |
| $\hat{\sigma}_r^2$ | 28.0174 *** | 28.446 *** | 28.342 *** |
| $\hat{\sigma}_\varepsilon^2$ | 24.9741 *** | 24.746 *** | 24.761 *** |
| $\hat{\rho}$ | 0.540 *** | 0.540 *** | 0.541 *** |
| Fit index |  |  |  |
| AIC | 48,246.1 | 48,251.6 | 48,255.3 |
| BIC | 48,253.1 | 48,258.7 | 48,262.3 |

The number of observations is 8167. *CTA* = centered Log(total assets); *CSG* = centered sales growth; *CTime* = centered time; *CCEOD* = centered CEO duality; *CPID* = centered percentage of independent directors; *CRQ* = centered regulatory quality; *CRL* = centered rule of law. Model 1 is the three-level model given in Equation (1). Model 2 is the three-level model given in Equation (3). Model 3 is the three-level model given in Equation (4). The tests associated with *CCEOD* and *CPID* in Model 1 and those associated with the cross-product terms in Models 2 and 3 are one-tailed. + $p < 0.10$; * $p < 0.05$; ** $p < 0.01$; *** $p < 0.001$.

*4.2. Robustness Check*

A robustness check was conducted by using rule of law instead of regulatory quality. The results are reported in Model 3 (Equation (4)) in Table 3. It was found that the moderating effect of rule of law on the relationship between CEO duality and ROA was significantly positive ($\hat{\gamma}_{11} = 0.0029$, $p = 0.099$), and that on the relationship between the percentage of independent directors and ROA was significantly negative ($\hat{\gamma}_{21} = -0.0186$, $p = 0.035$). The results with rule of law are consistent with those with regulatory quality.

**5. Discussion and Conclusions**

The purpose of this research is to examine the relationship between corporate governance and firm performance and how the relationship is moderated by country governance. Four hypotheses were developed and multi-level modeling was used for hypothesis testing with a dataset composed of 830 firms nested in 43 countries (out of the world's top 1000).

Hypotheses 1 that the effect of CEO duality is negative on firm performance was supported, consistent with previous studies (e.g., Dogan et al. 2013; Duru et al. 2016; Mubeen et al. 2020). Mubeen et al. (2020) find a negative relationship between CEO duality and firm performance. Tang (2017) shows that the effect of CEO duality is negative on firm performance in the United States. Furthermore, the agency theorists emphasize the negative relationship between CEO duality and firm performance. CEO duality may give CEOs enormous powers, and hence impact the efficacy of the board's governance.

Moreover, the findings indicating that more independent directors can lead to better firm performance support Hypotheses 2. The results agree with previous studies (i.e., Baysinger and Butler 1985; Peng 2004; Luan and Tang 2007; Liu et al. 2014; Masulis and Mobbs 2014; Kallamu 2016; Singh et al. 2018). Thus, the increase of independent directors on boards of firms indeed improves governance and performance of those firms.

Theoretically, if corporate governance is not adequate it needs to be uplifted to avoid adverse effects. According to the compensation theory, corporate governance compensates

for the deficiency of country governance. The results indicate that country governance moderates the effect of corporate governance on firm performance in such a way that the moderating effect is positive on the negative relationship between CEO duality and firm performance and is negative on the positive relationship between the percentage of independent directors and firm performance. Therefore, Hypotheses 3 and 4 are both supported. The results confirmed the findings of Bruno and Claessens (2010). The two research objectives were achieved. In sum, there exists a positive influence of board independence on financial performance, and the positive influence is negatively moderated by regulatory quality and rule of law, two indicators of country governance. Implementation of good corporate governance can be more effective when country governance is less adequate.

Previous studies have concluded that good firm performance is often due to good corporate governance practices. Analyzing the change in corporate governance is important for firm valuation (Morey et al. 2009). The Indian government is implementing good corporate governance in companies to improve firm performance (Arora and Sharma 2016). In New Zealand, Hossain et al. (2020) found that good corporate governance can enhance firm performance. Huang (2010) showed that a corporate governance model has a significantly positive impact on financial performance in Taiwan. Based on agency theory, this article further examined the effects of board independence, the core of corporate governance, on firm performance, and obtained the implications as follows: Avoidance of CEO duality and maintenance of a high proportion of independent directors can be golden rules for devising good governance mechanisms. Further, companies with no CEO duality and a high proportion of independent directors are good investment targets.

Based on compensation theory, this study expands our understanding of the relationships among corporate governance, country governance, and firm performance beyond Bruno and Claessens (2010). Country governance negatively moderates the positive effects of corporate governance on firm performance. It is implied that corporate governance acts as a driver for higher performance, particularly when country governance is inadequate. Corporate governance is particularly useful in this situation. Therefore, when poor country governance is encountered, firms need to pay more attention to corporate governance practices to improve their performance. Results also indicate that good country governance tends to reduce the positive influence of corporate governance on firm performance.

There exist research limitations in this study. First, only ROA was adopted as the indicator of firm performance. Second, the sample used consists of the world's top 1000 companies. Although they are representative, there may exist selection bias. Third, only CEO duality and the percentage of independent directors were considered as corporate governance indicators and regulatory quality and the rule of law as country governance indicators. More elements of corporate governance (such as ownership structure and concentration) and more elements of country governance (such as voice and accountability, political stability and absence of violence/terrorism, government effectiveness, and control of corruption) can be included in the multilevel model to gain more insight about their relationships, and these deserve future research.

**Funding:** The research was supported by grant MOST 107-2410-H-128-016 from the Ministry of Science and Technology, R.O.C.

**Institutional Review Board Statement:** Not applicable.

**Informed Consent Statement:** Not applicable.

**Data Availability Statement:** The source of data supporting reported results are shown in Table 1.

**Conflicts of Interest:** The author declares no conflict of interest.

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
