# Peer review of "On the Moderating Effects of Country Governance on the Relationships between Corporate Governance and Firm Performance"

_jrfm, doi:10.3390/jrfm14030140_

Round 1
Reviewer 1 Report
The research carried out is very interesting and relevant. The structure of the document is adequate, the abstract contains all the relevant information, the introduction exposes all the relevant information for the reader, the review of the literature is very deep and extensive, the methodology is correctly exposed and is adequate to achieve the stated objective , there is an adequate discussion of the results obtained.
The recommendations and suggestions are:
1.- Follow the template and authors guide. The document does not have the presentation of the magazine.
2.- After the discussion section, a new section "Conclusions" should be included, in which the conclusions obtained and their contribution to the theoretical field and practical implications are briefly exposed. Research limitations should also be included.
I congratulate the authors for their research
Author Response
Response to Reviewer 1 Comments
Point 1: Follow the template and authors guide. The document does not have the presentation of the magazine.
Response 1: I have revised the manuscript according to journal formats.
Point 2: After the discussion section, a new section "Conclusions" should be included, in which the conclusions obtained and their contribution to the theoretical field and practical implications are briefly exposed. Research limitations should also be included.
Response 2: I have renamed section “Discussion” to be “Discussion and Conclusion”. I have briefly exposed my contribution to the theoretical field and practical implications on lines 345-368 as follows:
Previous studies have concluded that good firm performance is often due to good corporate governance practices. Analyzing the change in corporate governance is important for firm valuation (Morey et al. 2009). The India government is implementing good corporate governance in companies to improve firm performance (Arora and Sharma 2016). In New Zealand, Hossain, Cahan and Adams (2000) found that good corporate governance can enhance firm performance. Huang (2010) showed that a corporate governance model has a significantly positive impact on financial performance in Taiwan. Based on agency theory, this article has further examined the effects of board independence, the core of corporate governance, on firm performance, and obtained the implications as follows: Avoidance of CEO duality and maintenance of a high proportion of independent directors can be golden rules for devising good governance mechanisms. Further, companies with no CEO duality and high proportion of independent directors are good investment targets.
Based on compensation theory, this study expands our understanding of the relationships among corporate governance, country governance and firm performance beyond Bruno and Claessens (2010). Country governance negatively moderates the positive effects of corporate governance on firm performance. It is implied that corporate governance acts as a driver for higher performance, particularly when country governance is inadequate. Corporate governance is particularly useful under the situation . Therefore, when poor country governance is encountered, firms need to pay more attention to corporate governance practices to improve their performance. Results also indicate that good country governance tends to reduce the positive influence of corporate governance on firm performance.
I have added research limitations on lines 369-375 as follows:
In this study, only ROA was adopted as the indicator of firm performance, and is a research limitation. Another limitation is that only CEO duality and the percentage of independent directors were considered as corporate governance indicators and regulatory quality and the rule of law as country governance indicators. More elements of corporate governance (such as ownership structure and concentration) and more elements of country governance (such as voice and accountability, political stability and absence of violence/terrorism, government effectiveness, and control of corruption) can be included in the multilevel model to gain more insight about their relationships, and these deserve future research.
Reviewer 2 Report
Thank you dear author for the research. I have gone through the complete paper and understood that you wanted to make sure, the relationship between corporate governance and the financial performance of firms. I have the following suggestions:
1. It is necessary to revise the abstract.
2. Research questions and research goals are necessary for the line of research motivation.
3. I didn't find any theoretical discussion. The study should incorporate a corporate governance theory to best explain the hypothesis.
4. It is necessary to update the literature review. Most of the references are before 2016. Only one reference is from 2020. Therefore it is necessary to update it.
5. My main concern is data and methods. Based on the recognized database why you have very minimum representation I am not sure. The highest representation data is from the USA. Why this is happened?
6. You have two hypotheses and two moderators but there are no control variables except asset. The prior study used many control variables. Corporate governance and financial performance both could be controlled by many things in the organization. So you need to add control variables.
7. You use three separate models. I didn't understand the meaning of that because you didn't use all variables in a model. So you need to draw a fourth model and putting all variables into a single regression. That will provide the exact regression results.
8. The dependent variable is very weak because only ROA is not enough to explain the financial performance of a firm at least you can use ROA and ROE both.
9. There is no robustness test to confirm the rigorousness of the study.
10. I have doubts about the implementation of the research because thousands of research have been done based on the same hypothesis. So what is your contribution?
I hope the authors will incorporate these issues and revise the paper.
Author Response
Response to Reviewer 2 Comments
Point 1: It is necessary to revise the abstract.
Response 1: The abstract has been carefully revised as follows:
Abstract: This study further empirically examines the influence of board independence on financial performance by using the world’s top 1000 firms. CEO duality and the percentage of independent directors are used as the indicators of board independence. Moreover, this study re-addresses the findings of Bruno and Claessen (2010) by giving supplement in theory and conducting tests for the influence of board independence on firm performance as well as the moderating effects of country governance, focusing on regulatory quality and rule of law, with multi-level modeling, a more sophisticated statistical approach. Four hypotheses, based on agency theory and compensation theory, were developed. The results indicated that CEO duality and the percentage of independent directors exerted, respectively, negative and positive influence on ROA, a firm performance indicator. Furthermore, regulatory quality and the rule of law positively moderated the negative effects of the former and negatively moderated the positive effects of the latter. Some practical implications were discussed based on the results obtained.
Point 2: Research questions and research goals are necessary for the line of research motivation.
Response 2: Research questions and research goals have been revised as follows:
Specifically, there are two research objectives of this study. First, this study further empirically examines the influence of board independence on financial performance by using the world’s top 1000 firms. CEO duality (meaning that the CEO plays the role of board chair also) and the percentage of independent directors are used as the indicators of board independence. Second, this study re-addresses the findings of Bruno and Claessen (2010) by giving supplement in theory and conducting tests for the influence of board independence on firm performance as well as the moderating effects of country governance, focusing on regulatory quality and rule of law, with multi-level modeling, a more sophisticated statistical approach.
Point 3: I didn't find any theoretical discussion. The study should incorporate a corporate governance theory to best explain the hypothesis.
Response 3: Hypotheses 1 and 2 are based on agency theory. Hypotheses 3 and 4 are based on compensation theory. More discussions have been given in the revised manuscript.
Point 4: It is necessary to update the literature review. Most of the references are before 2016. Only one reference is from 2020. Therefore it is necessary to update it.
Response 4: I have updated the literature review by citing five more recent articles regarding corporate governance and country governance as follows:
Koji, Kojima, Bishnu Kumar Adhikary, and Le Tram 2020. Corporate governance and firm performance: A comparative analysis between listed family and non-family firms in Japan. Journal of Risk and Financial Management 13: 215-34.
Mubeen, Riaqa, Dongping Han, Jaffar Abbas, and Iftikhar Hussain. 2020. The effects of market competition, capital structure, and CEO duality on firm performance: a mediation analysis by incorporating the GMM model technique. Sustainability 12: 3480–97.
Mukaddam, Shaa'ista and Sibindi Athenia. 2020. Corporate governance quality and financial performance of retail firms: Evidence using South African data. Academy of Accounting and Financial Studies Journal 24: 1-15.
Singh, Satwinder, Naeem Tabassum, Tamer K. Darwish, and Georgios Batsakis. 2018. Corporate Governance and Tobin's Q as a Measure of Organizational Performance. British Journal of Management 29: 171–90.
Sun, Sunny Li, Mike W. Peng, Ruby P. Lee, and Weiqiang Tan. 2015. Institutional open access at home and outward internationalization. Journal of World Business 50: 234–46.
Point 5: My main concern is data and methods. Based on the recognized database why you have very minimum representation I am not sure. The highest representation data is from the USA. Why this is happened?
Response 5: Hypotheses were tested with a dataset composed of 830 companies nested in 43 countries (out of the world’s top 1000). We believe the world’s top 1000 companies are representative. US companies occupy the greatest percentage.
Point 6: You have two hypotheses and two moderators but there are no control variables except asset. The prior study used many control variables. Corporate governance and financial performance both could be controlled by many things in the organization. So you need to add control variables.
Response 6: In addition to total assets, I have added sales growth (one-year growth of sales) and time as control variables in the models.
Point 7: You use three separate models. I didn't understand the meaning of that because you didn't use all variables in a model. So you need to draw a fourth model and putting all variables into a single regression. That will provide the exact regression results.
Response 7: Model 1 was used to test for Hypotheses 1 and 2. Model 2 was used to test for Hypotheses 3 and 4 with the country governance indicator of regulatory quality (RQ). Model 2 was used again to test for Hypotheses 3 and 4 with the country governance indicator of rule of law (RL) for robustness check. Two indicators of country governance, RQ and RL, were used for the purpose of robustness check. They need to be placed separately rather than jointly in Model 2.
Point 8: The dependent variable is very weak because only ROA is not enough to explain the financial performance of a firm at least you can use ROA and ROE both.
Response 8: I have mentioned in the manuscript that “ROA and Tobin’s Q are widely used indicators of firm performance in corporate governance research (e.g., Clacher et al. 2008; Bruno and Claessens 2010; García-Meca et al. 2015). In this study, ROA was used as the proxy of firm performance.” (lines 210-213) Moreover, I have added a research limitation as follows: “In this study, only ROA was adopted as the indicator of firm performance, and is a research limitation.” (lines 369)
Point 9: There is no robustness test to confirm the rigorousness of the study.
Response 9: Robustness check has been added and achieved. Please see Section 4.2. Robustness Check on lines 301-307.
Point 10: I have doubts about the implementation of the research because thousands of research have been done based on the same hypothesis. So what is your contribution?
Response 10: I have briefly exposed my contribution to the theoretical field and practical implications on lines 345-368 as follows:
Previous studies have concluded that good firm performance is often due to good corporate governance practices. Analyzing the change in corporate governance is important for firm valuation (Morey et al. 2009). The India government is implementing good corporate governance in companies to improve firm performance (Arora and Sharma 2016). In New Zealand, Hossain, Cahan and Adams (2000) found that good corporate governance can enhance firm performance. Huang (2010) showed that a corporate governance model has a significantly positive impact on financial performance in Taiwan. Based on agency theory, this article has further examined the effects of board independence, the core of corporate governance, on firm performance, and obtained the implications as follows: Avoidance of CEO duality and maintenance of a high proportion of independent directors can be golden rules for devising good governance mechanisms. Further, companies with no CEO duality and high proportion of independent directors are good investment targets.
Based on compensation theory, this study expands our understanding of the relationships among corporate governance, country governance and firm performance beyond Bruno and Claessens (2010). Country governance negatively moderates the positive effects of corporate governance on firm performance. It is implied that corporate governance acts as a driver for higher performance, particularly when country governance is inadequate. Corporate governance is particularly useful under the situation . Therefore, when poor country governance is encountered, firms need to pay more attention to corporate governance practices to improve their performance. Results also indicate that good country governance tends to reduce the positive influence of corporate governance on firm performance.
Reviewer 3 Report
It is an interesting paper, but the references are old, some of them from 20-30 years ago, such 1978, 1985. There are many interesting and original papers published recently. Please review and updated the references.
The structure of the paper is adequate, and I appreciate the well-written abstract with relevant information for the reader. The introduction presents the topic and aims, and gives an overview of the paper.
The recommendations and suggestions are:
- To develop the section of “Conclusions” and to indicate the contribution to the study case with four hypotheses with interesting results.
- To update the references, some of them are very old, such 20-30 years ago. This is not relevant for the research.
The research is interesting for the corporate governance and firm performance.
Author Response
Response to Reviewer 3 Comments
Point 1: To develop the section of “Conclusions” and to indicate the contribution to the study case with four hypotheses with interesting results.
.
Response 1: I have renamed section “Discussion” to be “Discussion and Conclusion”. I have summarized four hypotheses with interesting results on lines 323-344. I have mentioned my contribution to the theoretical field and practical implications on lines 345-368.
Point 2: To update the references, some of them are very old, such 20-30 years ago. This is not relevant for the research.
Response 2: I have updated the references by adding five more recent articles regarding corporate governance and country governance as follows:
Koji, Kojima, Bishnu Kumar Adhikary, and Le Tram 2020. Corporate governance and firm performance: A comparative analysis between listed family and non-family firms in Japan. Journal of Risk and Financial Management 13: 215-34.
Mubeen, Riaqa, Dongping Han, Jaffar Abbas, and Iftikhar Hussain. 2020. The effects of market competition, capital structure, and CEO duality on firm performance: a mediation analysis by incorporating the GMM model technique. Sustainability 12: 3480–97.
Mukaddam, Shaa'ista and Sibindi Athenia. 2020. Corporate governance quality and financial performance of retail firms: Evidence using South African data. Academy of Accounting and Financial Studies Journal 24: 1-15.
Singh, Satwinder, Naeem Tabassum, Tamer K. Darwish, and Georgios Batsakis. 2018. Corporate Governance and Tobin's Q as a Measure of Organizational Performance. British Journal of Management 29: 171–90.
Sun, Sunny Li, Mike W. Peng, Ruby P. Lee, and Weiqiang Tan. 2015. Institutional open access at home and outward internationalization. Journal of World Business 50: 234–46.
The articles by Jensen and Meckling (1976), Fama (1980) and Fama and Jensen (1983) are important references for agency theory. The articles by Champoux (1978) and Staines (1980) are important references for compensation theory. Although they are old, they need to be kept.
Round 2
Reviewer 1 Report
The authors have correctly followed the recommendations and suggestions.
Author Response
Point 1: The authors have correctly followed the recommendations and suggestions.
Response 1: The author is grateful to the reviewer for useful recommendations and suggestions.
Reviewer 2 Report
Thank you dear author for your best efforts. Now the paper looking nice but still I am in doubt about the methodological issue.
I know data representation quality but my concern was the imbalance representation. The US represents the highest quantity that may cause sample selection baisedness because you are using both country and firm-level variables.
Another issue I raised; regression model. It is necessary to put all variables/hypotheses into a single regression model for a better understanding of the influence. [see Masud et al. 2019]
Still, the study failed to use appropriate control variables.
Masud, M.A.K; Bae, S; Javier, M; & Kim, J.D. (2019). Board Directors’ Expertise and Corporate Corruption Disclosure: The Moderating Role of Political Connections, Sustainability,11 (16), 4491; https://doi.org/10.3390/su11164491
Author Response
Point 1: I know data representation quality but my concern was the imbalance representation. The US represents the highest quantity that may cause sample selection baisedness because you are using both country and firm-level variables.
Response 1: I have added one more research limitation on lines 372-373 of the revised manuscript as follows:
The sample used are world’s top 1000 companies. Although they are representative, there may exist selection bias.
Point 2: Another issue I raised; regression model. It is necessary to put all variables/hypotheses into a single regression model for a better understanding of the influence. [see Masud et al. 2019]
Response 2: The three-level model (Model 1) was used to test for Hypotheses 1 and 2 (the main effects of CEO duality (CCEOD) and the percentage of independent directors (CPID)). The variables CCEOD and CPID associated with Hypotheses 1 and 2 were put into a single equation (Equation 1). The three-level model (Model 2/3) was used to test for Hypotheses 3 and 4 (the moderating effects of RQ/RL on the influences of CEO duality and the percentage of independent directors on ROA). Variables CCEOD, CPID, CCEOD x CRQ and CPID x CRQ were put into another single equation (Equation 3). Similarly, variables CCEOD, CPID, CCEOD x CRL and CPID x CRL were put into a single equation (Equation 4). The effects of CCEOD and CPI in Equation 3/4 are not main effects shown in Equation 1. Equation 1 and Equation 3/4 need to be presented separately. Using separate models to test for main effects and interaction (moderation) effects is popular in the literature (e.g., Coombs and Gilley, 2005).
I have emphasized that Model 1 was used to test for the main effects of CEO duality and the percentage of independent directors on line 250 of the revised manuscript and Models 3/4 were used to test for the moderating effects of RQ/RL on lines 271-272.
Reference:
Coombs, Joseph E., K. Matthew Gilley. 2005. Stakeholder management as a predictor of CEO compensation: main effects and interactions with financial performance. Strategic Management Journal 26: 827-840.
Point 3: Still, the study failed to use appropriate control variables.
Response 3: The control variables used in this study include total assets (standing for firm size), one-year growth of sales (standing for sales growth), and time. They were also included in Bruno and Claessens (2010) (although logarithm of sales instead of logarithm of total assets was used as a proxy of firm size). It appears from Tables 3 that these three control variables were all statistically significant, showing that they are appropriate. I have added on lines 221-223 the following statement:
The control variables used include firm size, sales growth, and time (Bruno and Claessens 2010).
Round 3
Reviewer 2 Report
Thank you for the reply. Through I have some disagreement on methodological issues but I think the paper deserve to publish here.
Author Response
Point 1: Thank you for the reply. Through I have some disagreement on methodological issues but I think the paper deserve to publish here.
Response 1: I am grateful to the reviewer for insightful comments and suggestions. I thank the reviewer for his/her kindness.